# Attention Score-Based Multi-Vision Transformer Technique for Plant Disease Classification

**DOI:** 10.3390/s25010270

**Published:** 2025-01-06

**Authors:** Eu-Tteum Baek

**Affiliations:** Department of AI & Big Data, Honam University, Gwangju 62399, Republic of Korea; geodo100@gmail.com

**Keywords:** attention mechanism, plant pathology, deep learning in agriculture, multi-modal disease detection, vision-based diagnosis

## Abstract

This study proposes an advanced plant disease classification framework leveraging the Attention Score-Based Multi-Vision Transformer (Multi-ViT) model. The framework introduces a novel attention mechanism to dynamically prioritize relevant features from multiple leaf images, overcoming the limitations of single-leaf-based diagnoses. Building on the Vision Transformer (ViT) architecture, the Multi-ViT model aggregates diverse feature representations by combining outputs from multiple ViTs, each capturing unique visual patterns. This approach allows for a holistic analysis of spatially distributed symptoms, crucial for accurately diagnosing diseases in trees. Extensive experiments conducted on apple, grape, and tomato leaf disease datasets demonstrate the model’s superior performance, achieving over 99% accuracy and significantly improving *F*1 scores compared to traditional methods such as ResNet, VGG, and MobileNet. These findings underscore the effectiveness of the proposed model for precise and reliable plant disease classification.

## 1. Introduction

Plant diseases pose significant challenges to agricultural productivity and food security worldwide. Leaf diseases, in particular, can cause substantial yield losses and economic damages if left unmanaged. The timely and accurate identification of these diseases is crucial for implementing effective control measures to mitigate their impact. Traditional methods of leaf disease diagnoses often rely on visual inspection by trained experts. This process can be labor-intensive, time-consuming, and prone to subjective interpretation [1,2,3]. Moreover, the increasing complexity and variability of disease symptoms present further challenges for accurate diagnoses using conventional approaches. Visual inspections are not only resource-intensive but also limited by the availability and expertise of human diagnosticians, which can vary significantly.

To address these challenges, researchers have increasingly turned to advanced computational techniques, particularly artificial intelligence (AI) and machine learning (ML) algorithms, to develop automated systems for leaf disease detection and classification. By leveraging computer vision and pattern recognition, these approaches analyze digital images of plant leaves to identify disease symptoms with high accuracy and efficiency. AI and ML algorithms can process large volumes of image data rapidly and consistently, providing scalable solutions for timely and reliable disease diagnoses.

### 1.1. Materials and Methods

Recent advances in deep learning, particularly the use of convolutional neural networks (CNNs) and Vision Transformers (ViTs), have significantly enhanced the accuracy of leaf disease classification [4,5,6]. CNNs excel at capturing spatial hierarchies within images, enabling them to automatically learn and extract intricate features from raw image data. ViTs introduce self-attention mechanisms that handle larger image contexts, allowing the model to discern subtle patterns associated with various diseases more effectively. By exploiting these powerful architectures, researchers can achieve higher diagnostic precision, thereby providing critical data for early intervention and management strategies.

Building upon these developments, our approach integrates multiple leaves as input and incorporates a novel attention score mechanism for calculating the importance between embeddings extracted by Multi-Vision Transformer models. This mechanism dynamically adjusts attention weights based on the input images, prioritizing those that offer the most relevant information for disease classification. Through this method, we aim to overcome the limitations associated with conventional single-leaf examinations and leverage the spatial distribution of symptoms across multiple leaves for a more robust and comprehensive diagnostic outcome.

### 1.2. Challenges in Tree Disease Diagnosis

However, diagnosing diseases solely from individual leaves is often impractical, especially for tree diseases with symptoms that may be localized. Relying only on a single leaf overlooks the possibility that certain parts of the plant may show signs of disease while others appear healthy. To address this issue, our proposed method uses multiple leaves as input, providing a more complete assessment of the plant’s health and increasing the overall accuracy of disease diagnoses.

In this paper, we present a novel approach that leverages deep learning algorithms and a specialized attention scoring technique. By examining multiple leaves and dynamically adjusting attention weights, this method offers a more comprehensive, reliable, and actionable strategy for diagnosing plant diseases, ultimately contributing to more effective agricultural management practices.

## 2. Related Works

Several methodologies have been explored, ranging from traditional image processing techniques to modern machine learning algorithms. Notable approaches include the utilization of Support Vector Machines (SVMs) for sugarcane leaf disease detection and severity estimation based on segmented spot images, as proposed by Ratnasari et al., achieving an accuracy of 80% [7]. Gavhale et al. present an approach for unhealthy region detection in citrus leaves using statistical GLCM and SVM, achieving a recognition rate of 96% with an SVMRBF classifier [8]. Zhang et al. introduce a genetic algorithm-supported SVM (GA-SVM) for maize leaf disease recognition, outperforming traditional SVM with an accuracy of 92.59% [9]. Furthermore, Padol and Yadav employ SVM classifiers for grape leaf disease detection, achieving an accuracy of 88.89% [10]. Parikh et al. propose a cascaded classifier system for disease detection and severity estimation in cotton plants from unconstrained images, achieving an accuracy of 82.5% [11]. Thilagavathi and Abirami explore the application of SIFT features and SVM for diagnosing guava leaf diseases with an accuracy of 84% [12]. Other notable works include Sureesha et al. utilizing the KNN classifier for paddy leaf disease recognition with an accuracy of 76.59% [13]. Finally, Vaishnnave et al. employ the KNN classifier and Random Forest, respectively, for groundnut leaf disease and tomato plant disease detection, achieving commendable accuracies [14]. These studies collectively illustrate the evolving landscape of plant disease detection methodologies, showcasing the advancement from traditional techniques to sophisticated machine learning algorithms.

Several studies have been conducted in the field of plant disease classification and identification using artificial intelligence techniques, particularly deep learning models. In 2016, Ishak et al. utilized an Artificial Neural Network (ANN) to achieve an accuracy of 90.3% in classifying leaf diseases, demonstrating the effectiveness of ANN in this domain [15]. Subsequent research by Grinblat et al. in 2016 introduced the use of convolutional neural networks (CNNs) for plant identification based on vein morphological patterns [16]. Their study demonstrated superior performance compared to traditional machine learning algorithms, achieving accuracies of 93% and 98.8% with different CNN configurations. Ghosal et al. proposed an explainable deep machine vision framework in 2018, utilizing CNNs for plant stress phenotyping [17]. With an accuracy of 94.13%, their framework enabled the identification of various plant stresses, including bacterial blight and nutrient deficiencies. In 2019, Militante and Gerardo employed various deep learning models, including VGGNet, to detect sugarcane diseases, achieving an accuracy of 95.40% [18]. Their study highlighted the efficacy of deep learning in accurately identifying diseased sugarcane plants. Further advancements were made by Sethy et al. in 2020, who combined CNNs (ResNet50) with Support Vector Machines (SVMs) for rice leaf disease identification, achieving an impressive *F*1 score of 98.38% [19]. Continuing the trend of innovation, Hassan et al. in 2021 explored the use of CNN-based transfer learning models for plant disease detection, achieving remarkable accuracy rates, with EfficientNetB0 reaching 99.56% accuracy [20]. Moreover, recent studies have focused on hybrid approaches, such as the integration of Vision Transformers (ViTs) with CNNs, as demonstrated by Thakur et al., who achieved high accuracies in plant disease classification using a hybrid ViT-CNN model [21].

## 3. ViT (Vision Transformer)

A Vision Transformer (ViT) is a type of neural network architecture designed for image recognition tasks. The principles of the Transformer model, originally developed for natural language processing (NLP), are applied to the field of computer vision. The Vision Transformer comprises several key components, including patch embedding, positional encoding, the Transformer encoder, and the classification head.

The process begins with patch embedding, where the input image is divided into fixed-size patches. Each patch is flattened and linearly projected into a fixed-dimensional embedding. Formally, given an input image $\{P_{1}$, $P_{2},\;\ldots ,\;P_{n}\}$, each patch $P_{i}$ is embedded into a vector $E\left(P_{i}\right)$. This transformation converts the 2D image data into a sequence of 1D token embeddings, preparing it for processing by the Transformer model. Positional encoding is then added to these patch embeddings to provide spatial information, ensuring that the model can recognize the relative positions of the patches within the original image.

Central to the ViT is the Transformer encoder, which consists of multiple layers of multi-head self-attention mechanisms and feed-forward networks. Self-attention allows the model to weigh the importance of each patch relative to the others. To achieve this, the mechanism involves three key components such as query *Q*, key *K*, and value *V* vectors. The self-attention mechanism calculates the attention scores by taking the dot product of the query *Q_i_* with all keys *K_j_* from other patches, followed by a softmax operation to obtain attention weights. These weights are then used to compute a weighted sum of the value vectors *V_j_*, which represents how much information from each patch should contribute to the current patch. The equation for the self-attention mechanism is as follows:(1)$$Attention\left(Q_{i},\;K_{j},\;V_{j}\right)=Softmax\left(\frac{Q_{i}K_{j}^{T}}{\sqrt{d_{k}}}\right)V_{j}$$
where *d_k_* is the dimensionality of the key vectors, used to scale the dot product for numerical stability. This self-attention process captures complex relationships between patches, allowing the model to focus on relevant parts of the image. Each attention layer is followed by a position-wise feed-forward network, which is applied independently to each patch embedding. Residual connections and layer normalization are used to stabilize and improve the training process, ensuring efficient and effective learning of image representations. From this process, we obtain a 768-dimensional vector for each image, which is then exploited in the next stage for further tasks.

## 4. Attention Score-Based Multi-VIT

The proposed Attention Score-Based Multi-ViT model integrates multiple Vision Transformer (ViT) instances to enhance image classification performance. While ViTs capture global features of images using self-attention mechanisms, a single ViT may find it challenging to fully comprehend the complex characteristics of an image. This makes it difficult to diagnose a tree’s overall disease by examining just one leaf. Relying on individual pieces of information has limitations in grasping the overall state or patterns, so it is important to gather information from various perspectives. Therefore, we combine outputs from multiple pre-trained ViTs to learn diverse representations of the input image, extracting richer features in the process as shown in Figure 1.

Each ViT independently processes the input image and extracts a unique representation hi from the first (classification) token of its output, where *i* = 1, 2, …, *M*. These representations capture different features and patterns due to differences in each ViT’s architecture or initialization, allowing the model to understand the image from various perspectives.

To evaluate the importance of each ViT’s output, we introduce learnable attention weight vectors *W_i_*. The attention score ai is calculated through the dot product of the ViT’s representation hi and the attention weight *W_i_* as follows:(2)$$a_{i}=h_{i}^{T}W_{i}$$

The calculated attention scores are then normalized using the softmax function to produce normalized attention weights *α_i_*, representing the relative importance of each ViT. The normalization is performed as follows:(3)$$a_{i}=\frac{exp\left(a_{i}\right)}{\sum_{j=1}^{M}exp\left(a_{j}\right)}$$
where *M* = 5 is the number of ViTs used. The normalized attention weights *α_i_* reflect each ViT’s contribution to the final prediction. Using these normalized attention weights, we aggregate the individual ViT representations to form the final representation. The aggregation is performed through a weighted summation, as expressed in the following equation:(4)$$H=\sum_{i=1}^{M}a_{i}h_{i}$$

This aggregated representation integrates the diverse features obtained from multiple ViTs, more accurately reflecting the complex characteristics of the input image. To refine this combined representation, we apply a linear transformation layer, which can be mathematically described as
(5)$$H^{\prime }=Linear\left(H\right)$$

The refined representation *H′* is then passed to the classification head to generate the final logits.

The classification head typically consists of a fully connected layer with a softmax activation function for multi-class classification, allowing the model to output a probability distribution over predefined classes. This approach aims to improve the overall performance and generalization ability of the model by integrating diverse representations from multiple ViTs. By combining information obtained from various perspectives, the model can more effectively handle complex patterns and variations in images, which contributes to more accurate classification.

## 5. Experimental Results

This section presents a comprehensive quantitative evaluation of the proposed method compared to other models, utilizing metrics such as the *F*1 score, accuracy, mIoU, and confusion matrix. Information about the dataset and the input methods is also explained.

### 5.1. Dataset and Input Methods

To thoroughly assess the robustness of the proposed method, we utilized a variety of tree leaf datasets, including those related to apple trees and several other species. Each dataset comprises images of leaves exhibiting different disease conditions as well as healthy states. In the preprocessing stage, five leaf images with the same label were randomly selected from the dataset to train the proposed method. This approach ensures that the model learns accurate representations of each disease class from consistent input data.

During data composition, we further enhanced the dataset by systematically selecting four additional random images based on a sequential criterion ranging from 1 to n. This process was repeated iteratively, ensuring diverse combinations and providing the model with comprehensive learning opportunities. This iterative augmentation, performed over and over, generated a robust dataset, which was subsequently used for model training.

The input images were standardized to match the required format for the models. Specifically, images were resized to a suitable resolution compatible with the ViT architecture and other convolutional neural networks employed in the comparison. Data augmentation techniques, such as rotation, flipping, and scaling, were applied to enhance the diversity of the training data, thereby improving the model’s generalization capabilities.

### 5.2. Evaluation Metrics

This paper presents a comprehensive quantitative evaluation of the proposed method in comparison to other models, utilizing a range of metrics including the *F*1 score, accuracy, mean Intersection over Union (mIoU), and confusion matrix. The *F*1 score serves as a harmonic mean of precision and recall, offering a balanced evaluation when the dataset is imbalanced. It is defined by the equation
(6)$$F1=2\times \frac{Precision\times Recall}{Precision+Recall}$$

This metric is particularly useful in contexts where both false positives and false negatives carry significant costs, such as in disease detection tasks. A higher *F*1 score reflects the model’s ability to accurately balance the trade-offs between precision and recall, ensuring that both types of errors are minimized. Precision is defined as the ratio of true positive predictions to the total number of positive predictions made by the model:(7)$$Precision=\frac{TP}{TP+FP}$$

A high precision score indicates that the model is effective at minimizing false positives, making it particularly valuable in scenarios where incorrect positive predictions can lead to costly outcomes. Recall measures the model’s ability to correctly identify actual positive cases:(8)$$Recall=\frac{TP}{TP+FN}$$

High recall is essential in tasks where false negatives must be minimized, such as medical diagnoses, where failing to identify a positive case can have serious consequences. Accuracy provides an overall measure of the model’s performance, reflecting the proportion of correct predictions (both true positives and true negatives) across all cases:(9)$$Accuracy=\frac{TP+TN}{TP+TN+FP+FN}$$

Although accuracy is a widely used metric, it can be misleading in imbalanced datasets, where the model may achieve high accuracy by predominantly predicting the majority class correctly, while underperforming on the minority class. Therefore, accuracy should be used in conjunction with metrics like precision, recall, and the *F*1 score to provide a more holistic view of the model’s performance.

mIoU is commonly used in segmentation and object detection tasks to measure the overlap between the predicted region and the ground truth:(10)$$IoU=\frac{\left|Prediction\cap Ground\;Truth\right|}{\left|Prediction\cup Ground\;Truth\right|},\;\;\;mIoU=\frac{1}{C}\sum_{C=1}^{C}IoU_{C}$$
where *C* is the number of classes. A higher mIoU indicates a more accurate localization of diseased regions—an essential aspect for plant disease diagnoses.

To further understand the model’s performance, we utilize the confusion matrix, which breaks down the predictions into four categories: true positives (TPs), true negatives (TNs), false positives (FPs), and false negatives (FNs). This matrix helps identify specific areas where the model excels or requires improvement, such as reducing false positives (Type I errors) or false negatives (Type II errors). The confusion matrix also serves as the foundation for calculating precision, recall, and accuracy, offering a deeper insight into the types of errors the model is making and how they impact the overall performance.

### 5.3. Performance Evaluation

The experimental results reveal several key insights into the effectiveness of the proposed Attention Score-Based Multi-ViT model for plant disease classification, particularly in terms of accuracy and *F*1 scores. Across all datasets, the proposed model consistently outperforms traditional models, as evidenced by the figures in Table 1 and Table 2, demonstrating its ability to capture complex patterns in leaf images. This suggests that the integration of multiple ViT instances, each contributing diverse feature representations, significantly enhances classification performance. The introduction of attention score mechanisms further refines this process, allowing the model to prioritize the most informative image regions, resulting in fewer misclassifications, which can be observed in the confusion matrices presented in Figure 2.

The proposed model achieved remarkable accuracy across different plant diseases, notably with apple and grape leaf diseases, where it reached accuracies of 99.116% and 99.49%, respectively, as shown in Table 1 and Table 2. These figures surpass the performance of other models, underscoring the robustness of the Multi-ViT approach. ViT, ResNet50, and MobileNetV2, which are commonly used in image classification tasks, consistently underperformed compared to the proposed model, particularly in handling more subtle or diverse disease symptoms, as illustrated by the confusion matrices in Figure 3.

One key observation from these results is the significant improvement in *F*1 scores, particularly in datasets with overlapping symptoms or complex disease patterns, such as apple and grape leaf diseases. The attention score mechanism enables the model to dynamically weigh the importance of different ViT outputs, leading to more nuanced and precise classification decisions. This is clearly illustrated by the reduced misclassification rates, especially for classes that are traditionally harder to distinguish due to visual similarity, as shown in Figure 2 and Figure 3.

While the proposed model maintained competitive performance in tomato leaf disease classification, with an accuracy of 96.6924% and an *F*1 score of 0.9672, as demonstrated in Table 3, the results show a slight performance dip compared to its performance in other datasets. This trend likely reflects the complexity of tomato leaf disease symptoms, which might be more varied or less visually distinguishable. Nonetheless, the proposed model still outperformed the ViT and ResNet50 models, which struggled significantly in this task, with ViT achieving only 80.1963% accuracy as indicated in Table 3. The confusion matrices in Figure 4 highlight that, while MobileNetV2 and VGG performed well, the proposed model consistently reduced error rates across classes.

The confusion matrices presented in Figure 2, Figure 3 and Figure 4 further illustrate the superior generalization ability of the proposed model. By aggregating information from multiple ViT outputs, the model successfully minimizes misclassifications, particularly in classes that other models, such as ResNet50 and ViT, tend to confuse. This improvement in handling subtle intra-class variations across multiple leaf images underscores the model’s effectiveness in capturing the fine-grained details necessary for accurate disease diagnoses.

Additional experiments focusing on an ROC-AUC analysis were conducted for multiple models across different datasets as shown in Figure 5. The proposed model achieves the highest averaged ROC-AUC score, demonstrating superior performance in multi-class classification. ViT, ResNet50, and VGG show strong but slightly lower ROC-AUC values.

## 6. Discussion

The experimental results demonstrate that the proposed Attention Score-Based Multi-ViT model consistently outperforms traditional models, such as ResNet50, MobileNetV2, and standalone Vision Transformers, in plant disease classification tasks. For instance, the proposed model achieved an accuracy of 99.49% and an *F*1 score of 0.9949 in grape leaf disease classification, significantly surpassing ResNet50, which only achieved an accuracy of 96.73%. However, our model demonstrates additional advantages by effectively handling spatially distributed symptoms through a multi-leaf analysis. The integration of multiple ViTs with a dynamic attention score mechanism addresses the limitations of single-leaf-based diagnoses. By aggregating diverse feature representations, the model captures subtle intra-class variations, enabling more precise classification. This capability is particularly evident in apple leaf disease classification, where overlapping symptoms among classes traditionally cause high misclassification rates in other models. The confusion matrices presented in Figure 2, Figure 3 and Figure 4 illustrate a significant reduction in misclassification rates compared to competing models. From a practical perspective, the proposed model’s ability to process multiple leaves simultaneously enhances its applicability in real-world scenarios. Field conditions often involve complex symptom distributions across various leaves, making single-leaf-based models less effective. By accounting for the spatial distribution of symptoms, the Multi-ViT approach offers a more reliable solution for diagnosing tree diseases in field environments. Nonetheless, certain limitations remain. For example, the model exhibited a slight performance dip in tomato leaf disease classification, likely due to the higher variability in symptom presentation. Additionally, our experiments with YOLOv8 indicate that object detection-based models can also achieve strong performance in plant disease classification. However, the proposed Multi-ViT model demonstrates unique advantages in multi-leaf-based scenarios, offering more effective classification for complex disease patterns. Future work may explore hybrid approaches that combine the strengths of detection-based methods with multi-leaf strategies to further improve diagnosis accuracy. Future work may focus on incorporating environmental and contextual data to further enhance robustness and generalization.

## 7. Conclusions

In conclusion, the proposed Attention Score-Based Multi-ViT model demonstrates significant advancements in plant disease classification, consistently outperforming traditional models like ViT, ResNet50, VGG, YOLOv8, and MobileNetV2 in terms of accuracy, mIoU, and *F*1 scores. The integration of multiple ViTs, coupled with a dynamic attention score mechanism, enhances the model’s ability to capture complex patterns and prioritize relevant image features, leading to fewer misclassifications and superior generalization across different datasets. This was particularly evident in the classification of apple and grape leaf diseases, where the model achieved exceptional performance. While the model experienced a slight performance dip in the tomato leaf disease dataset, it still surpassed competing models, highlighting its robustness in handling varied and complex disease symptoms. The results suggest that this Multi-ViT approach offers a scalable and efficient solution for real-world agricultural applications, enabling more accurate and reliable plant disease diagnoses.

## Figures and Tables

**Figure 1 sensors-25-00270-f001:**
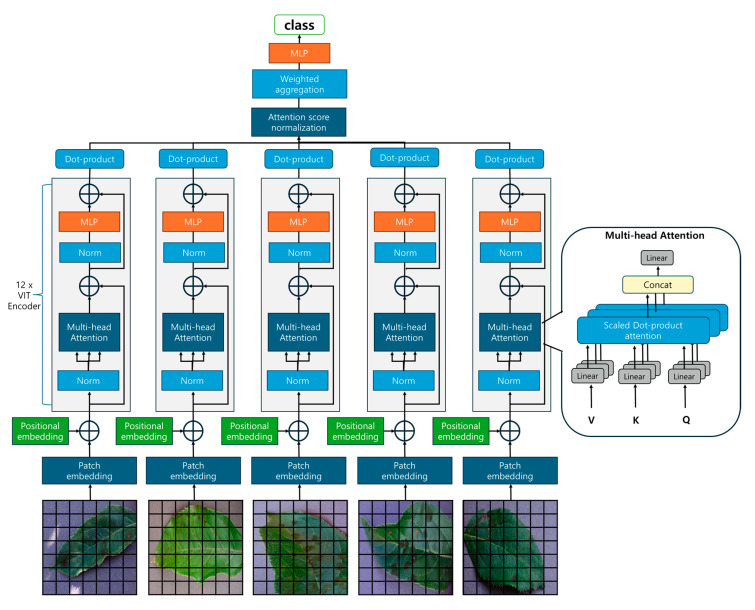
Model architecture for Attention Score-Based Multi-ViT for plant disease classification.

**Figure 2 sensors-25-00270-f002:**
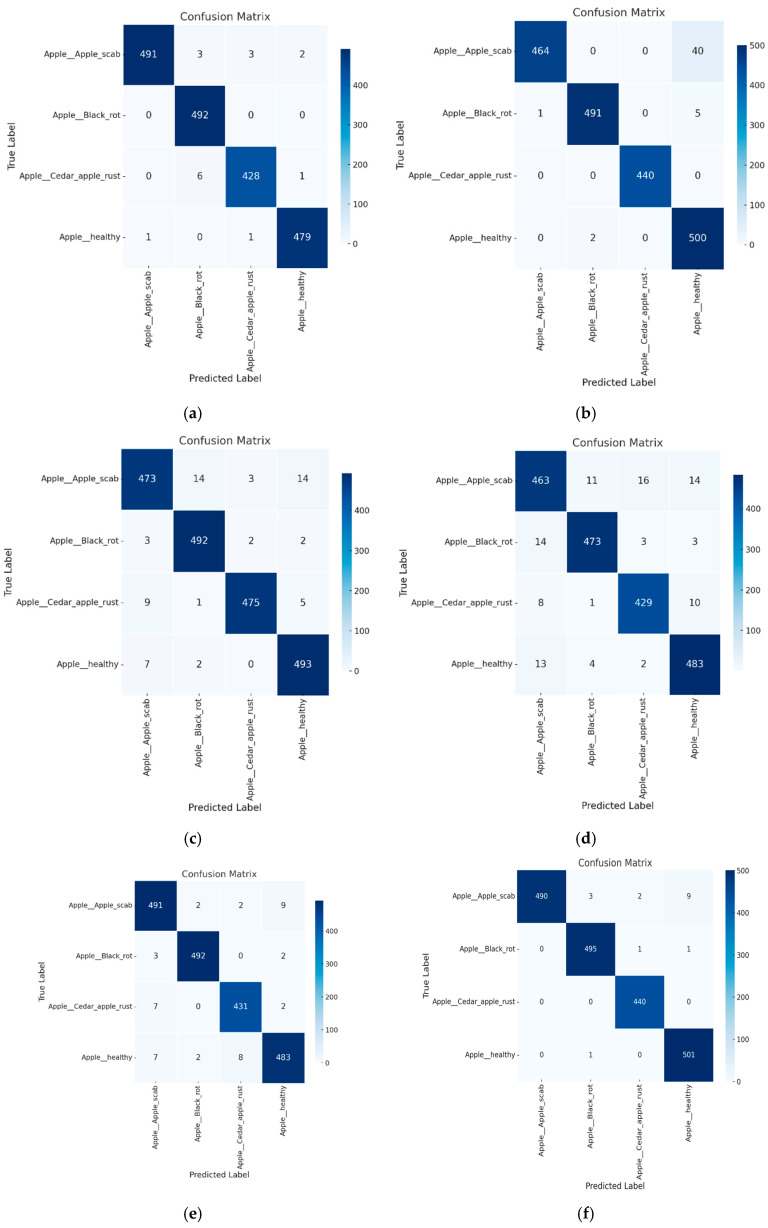
Confusion matrix for apple leaf disease classification across models. (**a**) Proposed; (**b**) VIT; (**c**) Resnet50; (**d**) VGG; (**e**) Mobilenet; (**f**) YOLOv8.

**Figure 3 sensors-25-00270-f003:**
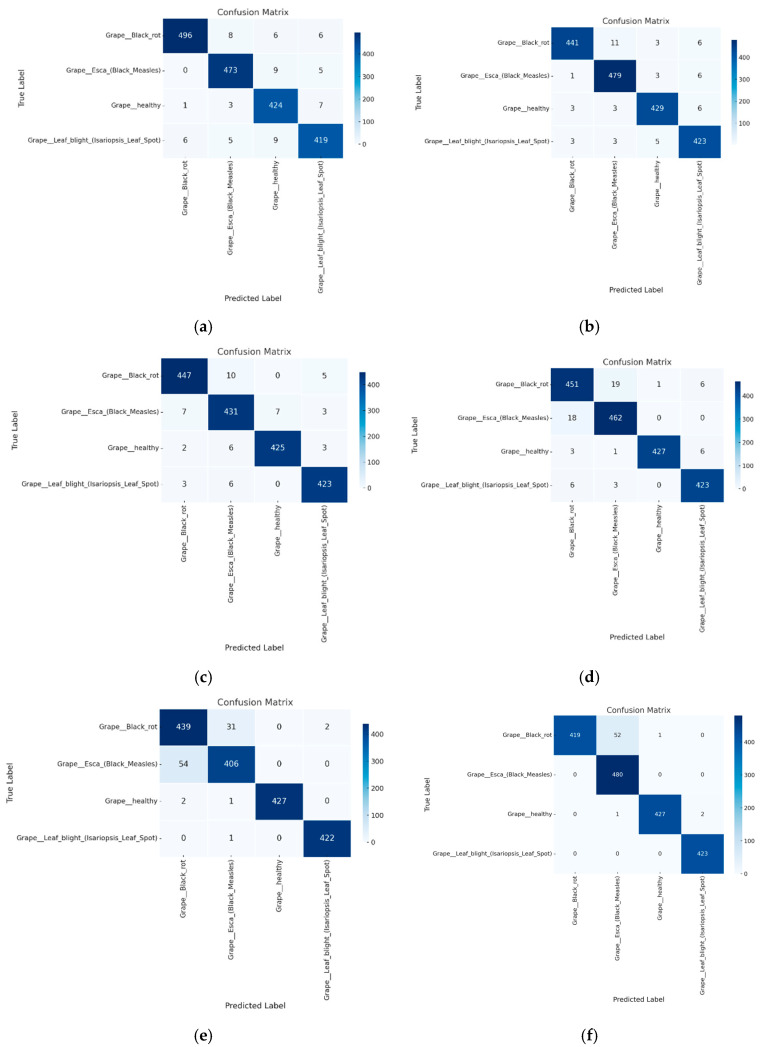
Confusion matrix for grape leaf disease classification across models. (**a**) Proposed; (**b**) VIT; (**c**) Resnet50; (**d**) VGG; (**e**) Mobilenet; (**f**) YOLOv8.

**Figure 4 sensors-25-00270-f004:**
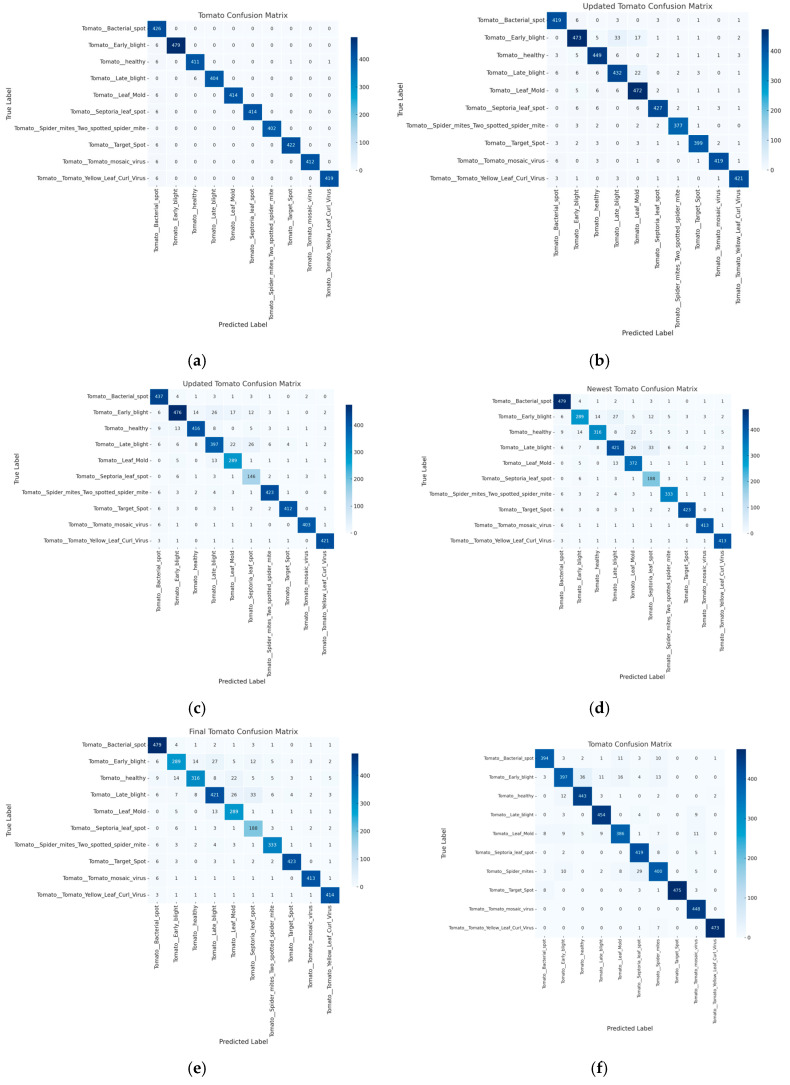
Confusion matrix for tomato leaf disease classification across models. (**a**) Proposed; (**b**) VIT; (**c**) Resnet50; (**d**) VGG; (**e**) Mobilenet; (**f**) YOLOv8.

**Figure 5 sensors-25-00270-f005:**
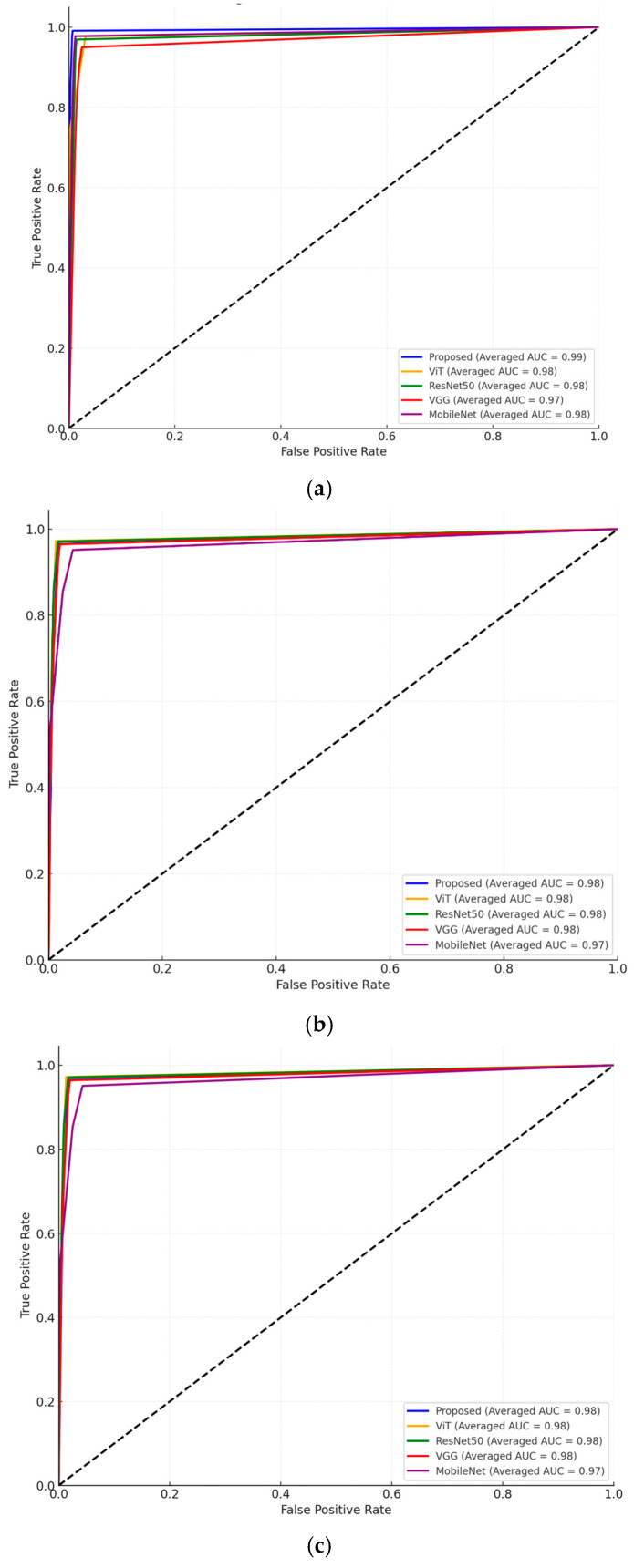
Comparative ROC-AUC curves for apple, grape, and tomato leaf disease classification across multiple models. (**a**) Apple leaf disease; (**b**) grape leaf disease; (**c**) tomato leaf disease.

**Table 1 sensors-25-00270-t001:** Performance Metrics of Different Models for Apple Leaf Disease Classification.

Method	Proposed	VIT [22]	Resnet50 [5]	VGG [23]	Mobilenet [24]	YOLOv8 [25]
*F*1 score	0.9911	0.9753	0.9690	0.9520	0.9850	0.99
Accuracy	99.1160	97.5242	96.9120	95.2136	98.5075	0.961
mIoU	0.982227	0.954214	0.94377	0.899457	0.967728	0.981025

**Table 2 sensors-25-00270-t002:** Performance Metrics of Different Models for Grape Leaf Disease Classification.

Method	Proposed	VIT	Resnet50	VGG	Mobilenet	YOLOv8
*F*1 score	0.9949	0.9822	0.9672	0.9496	0.9822	0.99
Accuracy	99.49	98.2271	96.7325	94.9585	98.2271	99.2
mIoU	0.9810	0.9360	0.937	0.904	0.9425	0.9373

**Table 3 sensors-25-00270-t003:** Performance Metrics of Different Models for Tomato Leaf Disease Classification.

Method	Proposed	VIT	YOLOv8	Resnet50	VGG	Mobilenet
*F*1 score	0.9672	0.7999	0.94	0.9221	0.9496	0.9283
Accuracy	96.6924	80.1963	93.5	92.2137	94.9585	92.8899
mIoU	0.9742	0.9647	0.9513	0.9728	0.9671	0.9565

## Data Availability

The data that support the findings of this study are available in the “New Plant Diseases Dataset” at Kaggle, accessible through https://www.kaggle.com/datasets/vipoooool/new-plant-diseases-dataset.

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
