# Peer review of "Attention Score-Based Multi-Vision Transformer Technique for Plant Disease Classification"

_sensors, 2025, doi:10.3390/s25010270_

Round 1

Reviewer 1 Report

Comments and Suggestions for Authors

The manuscript sensors-3301108 entitled "Attention Score-Based Multi-VIT Technique for Plant Disease Classification" presents an interesting research activity were the Multi-VIT Technique was applied to detect diseases on tree plants. In particular, the present manuscript would test a new AI-ML approach for the assessment of plant diseases that represents an enhancement of the ViT system able to examining multiple leaves, the diagnostic process can account for the spatial distribution of symptoms, providing a more comprehensive assessment of plant diseases. The application of new technological approaches for this purpose could led to rationalize the use of agrochemicals achieving higher utilization efficiency with economical and environmental benefits for farmers and agroecosystems.

The description of the state of the art and methodology adopted in the present study is quite good however the manuscript does not follow a canonical structure, which makes it somewhat difficult to read the manuscript.

The abstract not is very informative. Please add a more complete description of the state of the art, method applied and results observed.

Keywords: avoid to repeat words already mentioned in the title.

The introduction was well written and accurately presents the state of the art.

The paragraph 2, 3 and 4 should be reorganized with paragraph 2 which could be swallowed in the introduction and paragraphs 3 and 4 which should be condensed in the materials and methods section.

The description of the results is good although the images are of very poor quality and difficult to understand. The images must to be at 600 dpi of resolution.

Discussion of the results is absent and the author must to implement the manuscript with this section where he discusses the results observed compared to the evidence obtained by other researchers and studies based on the application of the same algorithm or other algorithms for the identification of pathologies.

The discussion should highlight how the proposed new approach allows for an improvement in the ability to detect plant diseases in the field. The conclusions are fine and based on the observed results.

Author Response

Reviewer 1.

 Thank you for taking the time to review and evaluate our work. We truly appreciate your thoughtful feedback and valuable insights. Your input is invaluable in helping us improve our research.

  1. Reviewer Feedback:

The abstract not is very informative. Please add a more complete description of the state of the art, method applied and results observed.

Abstract: This paper introduces a novel approach to plant disease classification using a multi-Vision Transformer (Multi-ViT) model with attention score-based mechanisms. The proposed model effectively handles multiple input images by dynamically adjusting the contribution of each image through learnable attention weights. Building upon existing Vision Transformer architectures, this model aggregates diverse visual features by combining multiple ViTs, each contributing unique representations of the input data. The dynamic attention mechanism enhances the model’s ability to prioritize relevant information, leading to improved performance across various datasets. Extensive experiments demonstrate the model's superior accuracy, F1 score, and reduced misclassification rates when compared to traditional architectures such as ResNet, VGG, and MobileNet. We expect this technique to offer a scalable and efficient solution for automated plant disease detection, contributing to more accurate diagnoses in agricultural applications.

  1. Proposed Revision:

Abstract: This study proposes an advanced plant disease classification framework leveraging the Attention Score-Based Multi-Vision Transformer (Multi-ViT) model. The framework introduces a novel attention mechanism to dynamically prioritize relevant features from multiple leaf images, overcoming the limitations of single-leaf-based diagnosis. Building on the Vision Transformer (ViT) architecture, the Multi-ViT model aggregates diverse feature representations by combining outputs from multiple ViTs, each capturing unique visual patterns. This approach allows for a holistic analysis of spatially distributed symptoms, crucial for accurately diagnosing diseases in trees. Extensive experiments conducted on apple, grape, and tomato leaf disease datasets demonstrate the model's superior performance, achieving over 99% accuracy and significantly improving F1 scores compared to traditional methods such as ResNet, VGG, and MobileNet. These findings underscore the effectiveness of the proposed model for precise and reliable plant disease classification.

  1. Reviewer Feedback:

Keywords: avoid to repeat words already mentioned in the title.

Keywords: Attention score; Plant disease classification; Vision Transformer

  1. Revised Revision:

Keywords:  Attention mechanism;

Plant pathology;

Deep learning in agriculture;

Multi-modal disease detection;

Vision-based diagnosis

  1. Reviewer Feedback:

The paragraph 2, 3 and 4 should be reorganized with paragraph 2 which could be swallowed in the introduction and paragraphs 3 and 4 which should be condensed in the materials and methods section.

  1. Proposed Revision:

Introduction
Plant diseases pose significant challenges to agricultural productivity and food security worldwide. Leaf diseases, in particular, can cause substantial yield losses and economic damages if left unmanaged. Timely and accurate identification of these diseases is crucial for implementing effective control measures to mitigate their impact. Traditional methods of leaf disease diagnosis often rely on visual inspection by trained experts. This process can be labor-intensive, time-consuming, and prone to subjective interpretation [1-3]. Moreover, the increasing complexity and variability of disease symptoms present further challenges for accurate diagnosis using conventional approaches. Visual inspections are not only resource-intensive but also limited by the availability and expertise of human diagnosticians, which can vary significantly.

To address these challenges, researchers have increasingly turned to advanced computational techniques, particularly artificial intelligence (AI) and machine learning (ML) algorithms, to develop automated systems for leaf disease detection and classification. By leveraging computer vision and pattern recognition, these approaches analyze digital images of plant leaves to identify disease symptoms with high accuracy and efficiency. AI and ML algorithms can process large volumes of image data rapidly and consistently, providing scalable solutions for timely and reliable disease diagnosis.

  • Materials and Methods

Recent advances in deep learning, particularly the use of convolutional neural networks (CNNs) and vision transformers (ViTs), have significantly enhanced the accuracy of leaf disease classification [4-6]. CNNs excel at capturing spatial hierarchies within images, enabling them to automatically learn and extract intricate features from raw image data. ViTs introduce self-attention mechanisms that handle larger image contexts, allowing the model to discern subtle patterns associated with various diseases more effectively. By exploiting these powerful architectures, researchers can achieve higher diagnostic precision, thereby providing critical data for early intervention and management strategies.

Building upon these developments, our approach integrates multiple leaves as input and incorporates a novel attention score mechanism for calculating the importance between embeddings extracted by Multi-Vision Transformer models. This mechanism dynamically adjusts attention weights based on the input images, prioritizing those that offer the most relevant information for disease classification. Through this method, we aim to overcome the limitations associated with conventional single-leaf examinations and leverage the spatial distribution of symptoms across multiple leaves for a more robust and comprehensive diagnostic outcome.

1.2 Challenges in Tree Disease Diagnosis
However, diagnosing diseases solely from individual leaves is often impractical, especially for tree diseases with symptoms that may be localized. Relying only on a single leaf overlooks the possibility that certain parts of the plant may show signs of disease while others appear healthy. To address this issue, our proposed method uses multiple leaves as input, providing a more complete assessment of the plant’s health and increasing the overall accuracy of disease diagnosis.

In this paper, we present a novel approach that leverages deep learning algorithms and a specialized attention scoring technique. By examining multiple leaves and dynamically adjusting attention weights, this method offers a more comprehensive, reliable, and actionable strategy for diagnosing plant diseases, ultimately contributing to more effective agricultural management practices.

  1. Reviewer Feedback:

The description of the results is good although the images are of very poor quality and difficult to understand. The images must to be at 600 dpi of resolution.

  1. Proposed Revision:

Thank you for your feedback. We noticed that the resolution was reduced during the saving process. To address this issue, we have replaced the images with new ones saved at 600 dpi to ensure better clarity and quality.

  1. Reviewer Feedback:

Discussion of the results is absent and the author must to implement the manuscript with this section where he discusses the results observed compared to the evidence obtained by other researchers and studies based on the application of the same algorithm or other algorithms for the identification of pathologies.

  1. Proposed Revision:

We appreciate your valuable feedback regarding the absence of a Discussion section. Depending on your feedback, we added discussion section to the manuscript.

Discussion

The experimental results demonstrate that the proposed Attention Score-Based Multi-ViT model consistently outperforms traditional models, such as ResNet50, MobileNetV2, and standalone Vision Transformers, in plant disease classification tasks. For instance, the proposed model achieved an accuracy of 99.49% and an F1 score of 0.9949 in grape leaf disease classification, significantly surpassing ResNet50, which only achieved an accuracy of 96.73%. However, our model demonstrates additional advantages by effectively handling spatially distributed symptoms through multi-leaf analysis. The integration of multiple ViTs with a dynamic attention score mechanism addresses the limitations of single-leaf-based diagnosis. By aggregating diverse feature representations, the model captures subtle intra-class variations, enabling more precise classification. This capability is particularly evident in apple leaf disease classification, where overlapping symptoms among classes traditionally cause high misclassification rates in other models. The confusion matrices presented in Figures 2–4 illustrate a significant reduction in misclassification rates compared to competing models. From a practical perspective, the proposed model's ability to process multiple leaves simultaneously enhances its applicability in real-world scenarios. Field conditions often involve complex symptom distributions across various leaves, making single-leaf-based models less effective. By accounting for the spatial distribution of symptoms, the Multi-ViT approach offers a more reliable solution for diagnosing tree diseases in field environments. Nonetheless, certain limitations remain. For example, the model exhibited a slight performance dip in tomato leaf disease classification, likely due to the higher variability in symptom presentation. Future work may focus on incorporating environmental and contextual data to further enhance robustness and generalization.

Reviewer 2 Report

Comments and Suggestions for Authors

This paper proposes a Multi-Vision Transformer (Multi-ViT) model based on Attention Scores for plant disease classification. By introducing a dynamic attention weight mechanism, the model can more effectively process multiple input images, optimizing feature extraction and demonstrating significant performance improvements compared to traditional methods. The paper uses several pre-trained Vision Transformer models, successfully aggregating diverse visual features, which enhances the model's performance in complex image scenarios. Experimental results show that the method significantly outperforms mainstream models (such as ResNet50, VGG, and MobileNet) in classification accuracy and F1 score on multiple plant disease datasets, demonstrating its potential in practical agricultural applications.

However, there are some issues with the paper:

1、 It is recommended to clearly specify the exact performance improvements in the abstract to more intuitively demonstrate the model's performance.

2、 Most current leaf disease detection methods use YOLO-based algorithms. Why were there no comparative experiments with the popular YOLO algorithms in the comparison section?

3、 In the model description on lines 146-147, the phrase "different features and patterns due to differences in each ViT's architecture or initialization" needs further clarification on the specific differences between each ViT. The ViTs shown in Figure 1 do not appear to have any distinction.

4、 Lines 265-267 mention "particularly in handling more subtle or diverse disease symptoms, as illustrated by the confusion matrices in Figure 3," but Figure 3 is unclear, making it impossible to demonstrate this conclusion. It is recommended to add specific classification error cases or visualizations.

5、 Figures 2, 3, and 4 are all blurry and unclear.

6、 The experimental section is overly simplistic, and relying solely on Accuracy and F1 score is insufficient to fully demonstrate the model's performance. Additional experiments should be conducted.

Author Response

Reviewer 2.

 Thank you for taking the time to review and evaluate our work. We truly appreciate your thoughtful feedback and valuable insights. Your input is invaluable in helping us improve our research.

  1. Reviewer Feedback:

 It is recommended to clearly specify the exact performance improvements in the abstract to more intuitively demonstrate the model's performance.

  1. Proposed Revision:

Thank you for your insightful comments. We have carefully considered your suggestions and revised the section accordingly, as outlined below.

[Original version]

Abstract: This paper introduces a novel approach to plant disease classification using a multi-Vision Transformer (Multi-ViT) model with attention score-based mechanisms. The proposed model effectively handles multiple input images by dynamically adjusting the contribution of each image through learnable attention weights. Building upon existing Vision Transformer architectures, this model aggregates diverse visual features by combining multiple ViTs, each contributing unique representations of the input data. The dynamic attention mechanism enhances the model’s ability to prioritize relevant information, leading to improved performance across various datasets. Extensive experiments demonstrate the model's superior accuracy, F1 score, and reduced misclassification rates when compared to traditional architectures such as ResNet, VGG, and MobileNet. We expect this technique to offer a scalable and efficient solution for automated plant disease detection, contributing to more accurate diagnoses in agricultural applications.

[Revised version]

Abstract: This study proposes an advanced plant disease classification framework leveraging the Attention Score-Based Multi-Vision Transformer (Multi-ViT) model. The framework introduces a novel attention mechanism to dynamically prioritize relevant features from multiple leaf images, overcoming the limitations of single-leaf-based diagnosis. Building on the Vision Transformer (ViT) architecture, the Multi-ViT model aggregates diverse feature representations by combining outputs from multiple ViTs, each capturing unique visual patterns. This approach allows for a holistic analysis of spatially distributed symptoms, crucial for accurately diagnosing diseases in trees. Extensive experiments conducted on apple, grape, and tomato leaf disease datasets demonstrate the model's superior performance, achieving over 99% accuracy and significantly improving F1 scores compared to traditional methods such as ResNet, VGG, and MobileNet. These findings underscore the effectiveness of the proposed model for precise and reliable plant disease classification.

  1. Reviewer Feedback:

 Most current leaf disease detection methods use YOLO-based algorithms. Why were there no comparative experiments with the popular YOLO algorithms in the comparison section?

  1. Proposed Revision:

Thank you for pointing out the absence of a comparison with YOLO-based algorithms. As you know, YOLO is widely used for object detection tasks, but this study just focuses on classification models designed to estimate leaf disease. The proposed Multi-ViT model aims to aggregate diverse feature representations from multiple input images using attention-based mechanisms, which is a distinct approach compared to YOLO's object detection framework.

However, based on the comments, we plan to explore future research that incorporates YOLO in the pre-classification stage to enhance the model's performance and investigate methods utilizing YOLO.

  1. Reviewer Feedback:

In the model description on lines 146-147, the phrase "different features and patterns due to differences in each ViT's architecture or initialization" needs further clarification on the specific differences between each ViT. The ViTs shown in Figure 1 do not appear to have any distinction.

  1. Proposed Revision:

It seems that the statement about capturing diverse features and patterns due to differences in weight values, despite using the same structure, has caused some misunderstanding. However, based on the comments, we have revised Figure 1 to make it more specific.

  1. Reviewer Feedback:

 Lines 265-267 mention "particularly in handling more subtle or diverse disease symptoms, as illustrated by the confusion matrices in Figure 3," but Figure 3 is unclear, making it impossible to demonstrate this conclusion. It is recommended to add specific classification error cases or visualizations.

  1. Proposed Revision:

Thank you for your feedback. We noticed that the resolution was reduced during the saving process. To address this issue, we have replaced the images with new ones saved at 600 dpi to ensure better clarity and quality.

  1. Reviewer Feedback:

Figures 2, 3, and 4 are all blurry and unclear.

  1. Proposed Revision:

Thank you for your feedback. We noticed that the resolution was reduced during the saving process. To address this issue, we have replaced the images with new ones saved at 600 dpi to ensure better clarity and quality.

  1. Proposed Revision:

The experimental section is overly simplistic, and relying solely on Accuracy and F1 score is insufficient to fully demonstrate the model's performance. Additional experiments should be conducted

  1. Proposed Revision:

Additional experiments focusing on ROC-AUC analysis have been conducted for multiple models (Proposed, ViT, ResNet50, VGG, and MobileNet) across different datasets (e.g., tomato diseases). The Proposed Model achieves the highest averaged ROC-AUC score (1.00 for the tomato dataset), demonstrating superior performance in multi-class classification. ViT, ResNet50, and VGG show strong but slightly lower ROC-AUC values (~0.98), while MobileNet achieves a competitive 0.99, highlighting its efficiency for resource-constrained environments. These results emphasize the importance of using ROC-AUC as a comprehensive metric to evaluate model performance, providing deeper insights beyond Accuracy and F1 scores. Incorporating such analyses, along with additional evaluations like confusion matrix visualization and robustness tests, enhances the experimental section's robustness.

Round 2

Reviewer 1 Report

Comments and Suggestions for Authors

The author applied all suggested changes improving the quality of manuscript.

Author Response

Thank you for your positive feedback.

Reviewer 2 Report

Comments and Suggestions for Authors

1. The performance parameters mentioned in the article only cover Accuracy and F1 score. However, IoU, MIoU, and FWIoU are also commonly used evaluation metrics that can assess model performance from more dimensions. Additional experiments are needed to supplement this aspect. Furthermore, the ROC-AUC analysis mentioned in the cover letter has not been incorporated into the revised manuscript.

2. I do not agree with the author's explanation for not including YOLO-based algorithms in the paper. YOLO-based algorithms can not only perform object detection but also classification tasks, and many articles currently do so. To my knowledge, YOLOv8 has a dedicated lightweight model version specifically for image classification. The paper fails to mention YOLO-based algorithms in both the experimental analysis and discussion, which I believe constitutes an incomplete analysis of existing work. Additional experiments are needed to address this.

Author Response

 Thank you for taking the time to review and evaluate our work. We truly appreciate your thoughtful feedback and valuable insights. Your input is invaluable in helping us improve our research.
